# Determination of Critical Transformation Temperatures for the Optimisation of Spring Steel Heat Treatment Processes

**Velaphi Jeffrey Matjeke** [1,*], **Josias Willem van der Merwe** [1] **and Nontuthuzelo Lindokuhle Vithi** [2]

1 School of Chemical and Metallurgical Engineering, University of the Witwatersrand, Johannesburg 2000, South Africa; Josias.VanDerMerwe@wits.ac.za
2 Transnet Engineering, Research Development, Private Bag X528, Kilnerpark 01272, South Africa; lindokuhle.vithi@transnet.net
* Correspondence: Velaphi.matjeke@transnet.net; Tel.: +27-(0)12-743-0515

**Abstract:** Bogie spring performance can be improved by using the exact heat treatment process parameters. The purpose of the study is to determine the critical transformation temperatures and investigate the effect of the cooling rates on microstructural and mechanical properties. The precise determination of the required cooling rates for the particular grade of steel is important in order to optimise the heat treatment process of heavy-duty compression helical spring manufacturing. A traditional heat treatment system for the manufacture of hot coiled springs requires heating the steel to homogenize austenite; then, it is decomposed to martensite by rapid cooling. By analyzing the transition properties by heating and differing cooling rates, this analysis examines the thermal behaviour of high strength spring steel. Using the dilatometer and differential scanning calorimeter, scanning electron microscope, optical microscope, and hardness checking, critical transition temperatures and cooling rates of three springs steels were measured. Although the thermal transformation of materials has been researched for decades using dilatometers, not all materials have been characterized. The research offers insights into the critical transformation temperatures for the defined grades of spring steel and the role of cooling rates in the material's properties. Mechanical properties are influenced by the transition data obtained from the dilatometric analysis.

**Keywords:** dilatometer; spring steel; critical transformation temperature





## 1. Introduction

The recent global economic development has led to an increased demand for rail freight, resulting in higher traffic, tonnage, and train speeds [1–3]. The durability of safety-critical suspension elements, such as bogie springs, must be strengthened in order to run heavy haul trains safely [3]. To bear the load within a specified range without plastic deformation, these springs depend on elastic deformation [4]. In determining the material applicability of the springs, the EN 10,089 norm relies on hardness and strength [5]. The chemical composition and cooling rate are usually a function of the microstructures, hardness, and strength of steel [6]. The steels chosen for this investigation were selected to maximize the development of bogie springs and are graded as grades 55Cr3, 54SiCr6, and 52CrMoV4 by EN 10089. With respect to alloying elements, these steel grades differ considerably. The 54SiCr6 grade has a significantly higher silicon content than the other two grades which, in theory, delays carbon diffusion within steel, thus promoting martensite formation at lower quenching rates, thereby improving hardenability [7]. Grades 55Cr3 and 52CrMoV4 have higher manganese contents and 52CrMoV4 has a higher chromium content than the other two, and has a significant increase in hardenability. The manufacturing process needs to be optimised in order to enhance the mechanical properties of hot coiled springs. A thorough understanding of the critical transformation temperatures and behavior of the spring steel grades selected is therefore required. Austenitisation and quenching processes are the key steps in the development of hot coiled springs [8].

The acceptable hardened microstructure for these types of springs is martensite, which is dependent on the lower critical temperature ($Ac_1$), upper critical temperature ($Ac_3$), and martensite start (Ms) temperature [9,10].

$Ac_1$ and $Ac_3$ are regarded as the beginning of austenite phase transformation and the end of austenite formation during heating. The empirical data generated from the JMatpro software show the critical transformation temperatures of steel grade 54SiCr6 to be significantly lower than the one predicted using Andrew's equation [11]. An earlier study also showed that there is no agreement between the JMatPro simulation software and Andrew's equation in determining the Ms temperature of 54SiCr6 steel grade [12]. Andrew's equation is commonly used to estimate the temperature of $Ac_1$, $Ac_3$, and the Ms temperature [13,14]. The equations are listed as Equations (1)–(3).

Thermal dilatometry can be used during heating and cooling procedures to precisely determine the critical transformation temperature [15,16]. This contributes to the existing knowledge gap. The critical transformation temperature can be accurately determined using dilatometry. The dilatometric phase change is accompanied by a significant change in length, along with a temperature increase [17]. The transformation of the martensite depends on the stabilization of the austenite process, which depends to a large extent on heating above $Ac_1$ and $Ac_3$ [18]. The unreliability in the Ms temperature has warranted for the heating and cooling characteristics of the three grades to be investigated.

The transformation kinetics of the steel phases are known and understood, but for the unique steel grades, this study aims to predict the optimum transformation characteristics. For instance, martensite is a diffusionless transformation that occurs by habit plane shear when the steel is rapidly cooled from the austenitic range [19]. The transformation encompasses the compression of the BCT lattice parameter c-axis and the extension of the a-axis. Carbon atoms are trapped within the BCT structure. Generally, a martensite microstructure is attained when high-strength spring steel is supercooled from the austenitic region [20,21]. It is often difficult to achieve 100 percent martensite on spring steel without causing quench cracks [22–24]. The material's ability to harden is largely influenced by the chemical composition and quenching capacity of the quenching media. As a result, the hardened microstructure usually has small percentages of either retained austenite, pearlite, or bainite. The purpose of this study was to determine the exact critical temperature of the transition and the effect of the cooling rate on the microstructure and mechanical properties of the spring steel. The selected spring steels are typically susceptible to quenching cracks and microstructural defects when the quenching process is not controlled; thus, it is necessary to determine the material's critical transformation temperature and behavior during cooling [22].

$$Ac_1(^{\circ}C) = 723 - 20.7(\%Mn) - 16.9(\%Ni) + 16.9(\%Cr) + 290(\%As) + 6.38(\%W); \quad (1)$$

$$A_3(^{\circ}C) = 910 - 203\left(\sqrt{\%C}\right) - 15.2(\%Ni) + 44.7(\%Si) + 104(\%V) + 31.5(\%Mo) + 13.1(W); \quad (2)$$

$$M_s(^{\circ}C) = 539 - 432(\%C) - 30.4(\%Mn) - 17.7(\%Ni) - 12.1(\%Cr) - 7.5(\%Mo) + 13.1(W) \quad (3)$$

## 2. Materials and Methods

To perform the thermal analysis and determine the effect of cooling speeds on microstructures and mechanical properties, the three selected spring steel grades were prepared for differential calorimeter scanning and dilatometry.

Dilatometer samples were machined to a length of 10 mm and a diameter of 5 mm, while the differential scanning calorimeter (DSC) samples were prepared toward smaller particles by grounding the solid-state metal. For each grade of spring steel, the dilatometric test samples were heated to 860 °C, followed by soaking them at the same temperature, before cooling them at different speeds. Helium gas was used as the cooling media. In order to produce dilatometric data for the steels, constant cooling speeds were used. Using thermal expansion against temperature, the critical transition temperatures were calculated by calculating the dilation, while the DSC used heat flow. The criteria for heat treatment are

outlined in Table 1. The chosen steel grades used in this experiment were 55Cr3, 54SiCr6, and 52CrMoV4. The chemical compositions were validated using a spectrometer (Bruker Q4 Tasman, Carteret, NJ, USA), as seen in Table 2.

**Table 1.** Dilatometry parameters used on the three materials.

| Temperature (°C) | Cooling Rate (°C/s) | Heating Rate (°C/s) | Soaking Time (h) |
|:---:|:---:|:---:|:---:|
| 860 | 0.01 | 2 | 45 |
| 860 | 0.1 | 2 | 45 |
| 860 | 1 | 2 | 45 |
| 860 | 10 | 2 | 45 |
| 860 | 20 | 2 | 45 |
| 860 | 30 | 2 | 45 |
| 860 | 100 | 2 | 45 |

**Table 2.** Chemical composition of the steels in wt%.

| Element | % C | % Si | % Mn | % P | % S | % Cr | % Ni | % Cu | % Mo | % V |
|:---|:---|:---|:---|:---|:---|:---|:---|:---|:---|:---|
| 55Cr3 | 0.58 | 0.32 | 0.92 | 0.013 | 0.004 | 0.79 | 0.06 | 0.10 | 0.01 | 0.01 |
| 54SiCr6 | 0.56 | 1.33 | 0.71 | 0.014 | 0.002 | 0.75 | 0.01 | 0.01 | 0.00 | 0.06 |
| 52CrMoV4 | 0.55 | 0.28 | 0.92 | 0.012 | 0.007 | 1.05 | 0.01 | 0.01 | 0.19 | 0.11 |

The heat-treated dilatometer test samples were mounted and prepared for microscopic inspection by grinding and polishing them to a 1-µm finish. Before the metallographic analysis using both the optical microscope (Olympus, Tokyo, Japan) and SEM, the samples were etched with 2 percent nital etchant. The general microstructure of the as-received steel rods from the mill was a pearlitic structure. To assess the hardness of the components compared to the microstructure, the samples were further subjected to micro-Vickers hardness testing. A micro-Vickers hardness test, with 0.5 kg.f, was used for microhardness; five indentations were taken per sample with 400-µm radius spacing between the indentations to avoid the deformation zones. The continuous cooling curves for the three steels were generated from the dilation curves. The fastest cooling rate of 100 °C/s was used to determine the Ms temperature of the three steels. The $Ac_1$, $Ac_3$, and Ms temperature were compared to the data produced by JMatPro software and DSC. The JMatPro simulation data and phase distribution were determined using the calculation of phase diagrams (CALPHAD) method as a function of the chemical composition of each steel [25]. The samples for the DSC tests were heated to 860 °C at 10 °C/s for the purpose of validating the $Ac_1$.

## 3. Results and Discussion

### 3.1. Critical Transformation Temperature

For the purpose of process optimisation, dilatometric and DSC experiments were performed to assess the precise phase transition temperatures. The dilation characteristics of the three steels are seen in Figure 1 and Table 3. A tangent line is drawn along the graph, and the temperature at which thermal expansion changes is called the critical transformation temperature.

The DSC measurements verified the dilatometer results. The actual thermal test measurements were compared to the JMatPro estimations and Andrew's equation results. It was found that there was no unanimity of the test results, between JMatpro, and Andrew's equation outputs. The optimisation of the heat treatment parameters is largely dependent on the $Ac_1$, $Ac_3$, and Ms temperature. A higher $Ac_1$ means a longer cooling period. It is evident from the results that a general approach would be ineffective to optimise the heat treatment parameters for the three steel grades. According to the dilation findings, it would take the least time for 55Cr3 to initiate martensite formation, followed by 52CrMoV4. It is, however, noteworthy that due to a higher Ms temperature, 54SiCr6 is the most hardenable.

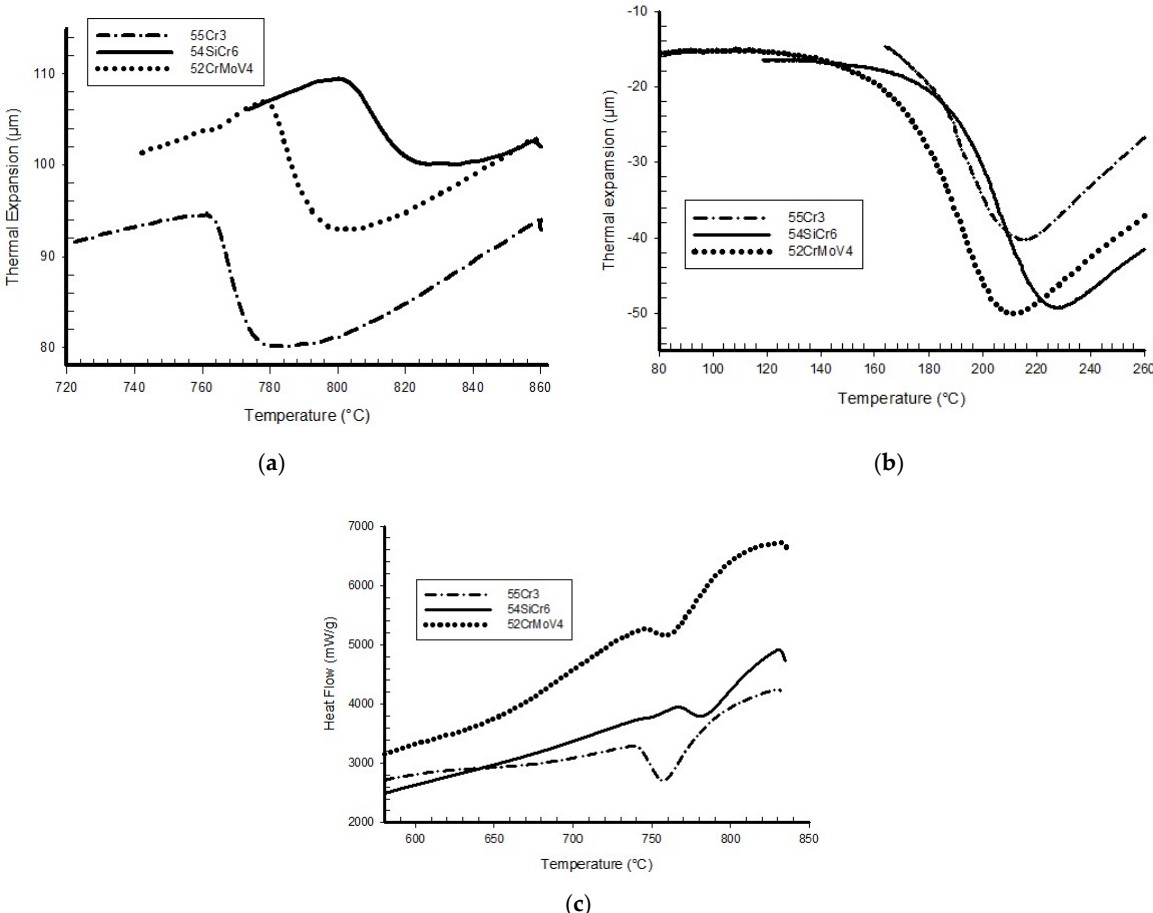

**Figure 1.** Critical transformation temperatures for the three steels: (**a**) $Ac_1$ and $Ac_3$ using dilatometer, (**b**) Ms temperature using dilatometer, and (**c**) DSC phase transition.

**Table 3.** Critical transformation temperatures.

| Identity | Andrew's Equation | | | Dilatometer | | | DSC | | JMatPro | | |
|---|---|---|---|---|---|---|---|---|---|---|---|
| Transition Temperature (°C) | $Ac_1$ | $Ac_3$ | Ms | $Ac_1$ | $Ac_3$ | Ms | $Ac_1$ | Ms | $Ac_1$ | $Ac_3$ | Ms |
| 55Cr3 | 726 | 740 | 255 | 753 | 818 | 218 | 751 | 225 | 738 | 741 | 253 |
| 54SiCr6 | 759 | 945 | 271 | 793 | 846 | 235 | 777 | 235 | 767 | 775 | 260 |
| 52CrMoV4 | 727 | 909 | 264 | 772 | 823 | 216 | 762 | 227 | 749 | 750 | 266 |

*3.2. The Effect of Soaking Time on Phase Transformation*

The austenite critical transformation temperature for 54SiCr6 steel is higher compared to 55Cr3 and 52CrMoV4; however, it takes less time for austenite to stabilise. The rule of thumb is that for complete austenitisation, the length of time required for steel to be homogeneous is 1 h for a 25-mm diameter [26]. Carbon and other alloying elements require time to be completely in solution [27]. It can be observed from Figure 2 that there is a change in the length of the different steels during isothermal treatment at 860 °C.

The change in length can be described as phase transformation from a BCC phase to a smaller FCC phase. This change in length stabilises over time; however, the rate of stabilisation varies from one steel to another. Figure 2 shows that the thermal expansion of 54SiCr6 was stabilised approximately 30 min after soaking, while 52CrMoV4 was not stabilised after fourty five (45) min of soaking. This proves that complete austenitisation does not only depend on time but also on composition. Therefore, the $Ac_1$ and $Ac_3$

temperatures and the austenite stabilisation time are to be carefully considered during the heat treatment optimisation process.

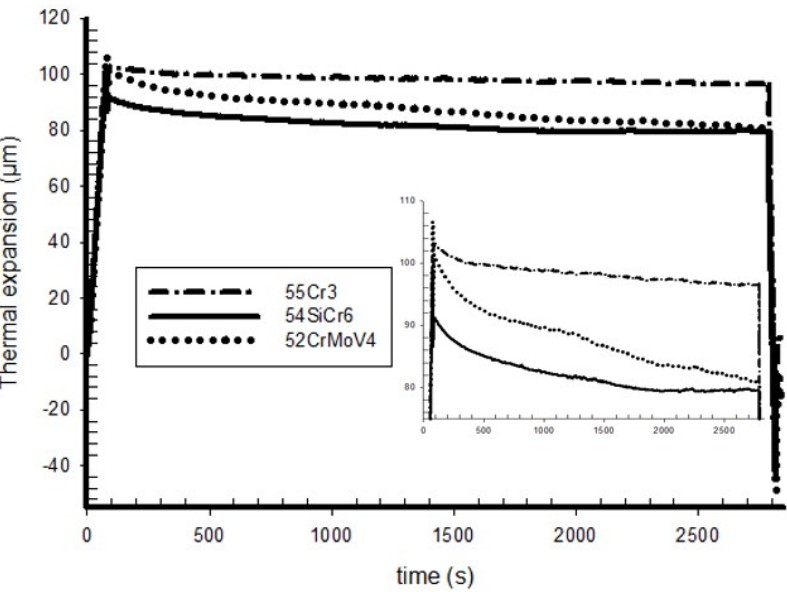

**Figure 2.** Thermal expansion rates for the three steels.

### 3.3. The Effect of Cooling Rates on Phase Transformation

The thermal dilatometric treatment results are presented in Figure 3. These graphs present the critical transformation temperature achieved by varying the constant cooling rates.

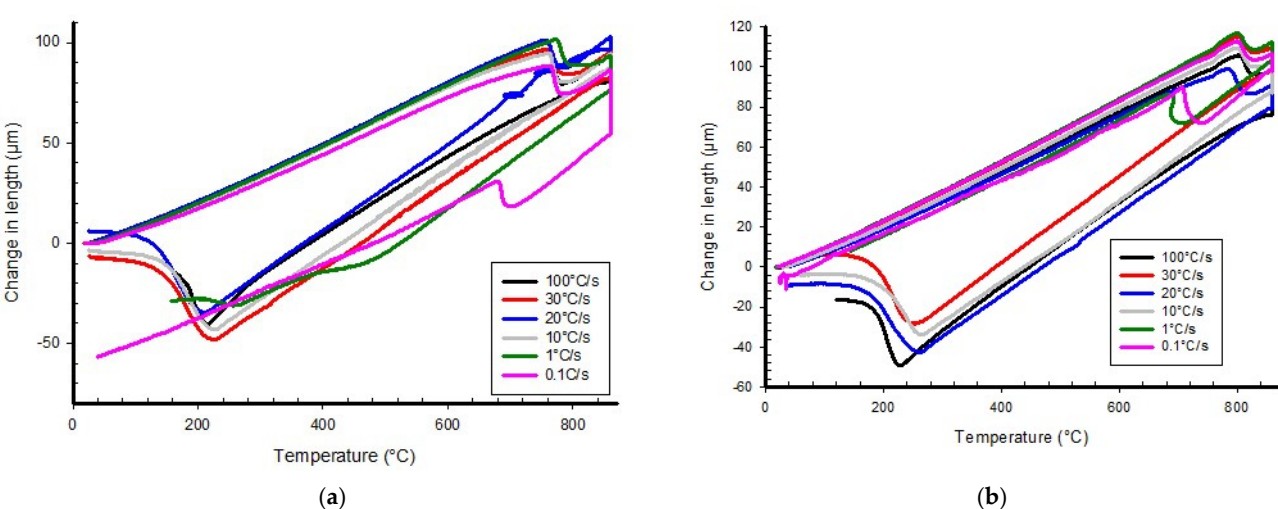

(**a**)  (**b**)

**Figure 3.** *Cont.*

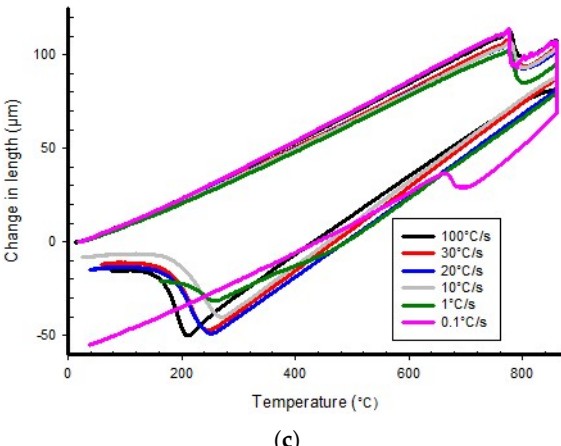

(**c**)

**Figure 3.** Dilatometric graphs of (**a**) 55Cr3, (**b**) 54SiCr6, and (**c**) 52CrMoV4 at various cooling rates.

There was a general increase in the Ms temperature for the three steels with a decrease in the cooling rates, which varied as shown in Figure 4.

The variation in the Ms temperature is attributed to chemical compositional change during cooling. Slower cooling rates allow for the diffusion process, whereby carbides may form, changing the overall composition of the steel. 54SiCr6 exhibits higher Ms temperature within the cooling rates of 20 °C to 100 °C/s. At 1 °C/s and lower, for 54SiCr6, the cooling rates were not significant to form martensite. This means that it was more hardenable than 55Cr3 and 52CrMoV4. Although 55Cr3 exhibited the lowest Ms temperature at 100 °C/s, it showed the highest Ms temperature at 1 °C/s. The Ms temperature of 55Cr3, at 100 °C/s, was comparable to that of 52CrMoV4. However, the significant increase in the Ms temperature of 52CrMoV4 was only observed with a cooling rate of 10 °C/s, before gradually decreasing at 1 °C/s cooling rate. 55Cr3 was stable between 100 °C/s and 10 °C/s before showing an increase in Ms temperature at 1 °C/s. At 0.1 °C/s, none of the materials formed martensite. All the permutations stated above must be taken into consideration when optimising the heat treatment parameters. The microstructures of each steel grade are presented in Figures 5–7.

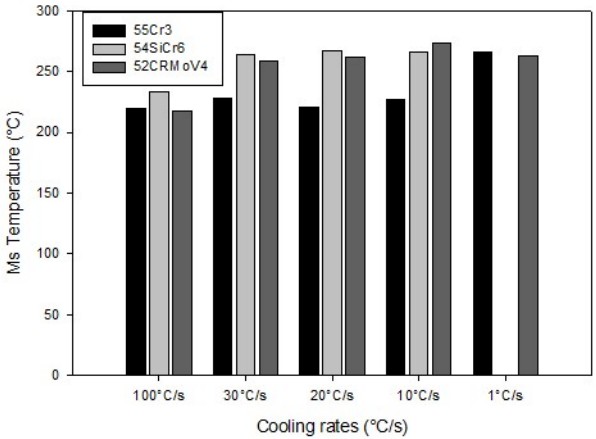

**Figure 4.** The effect of cooling rates on Ms temperature.

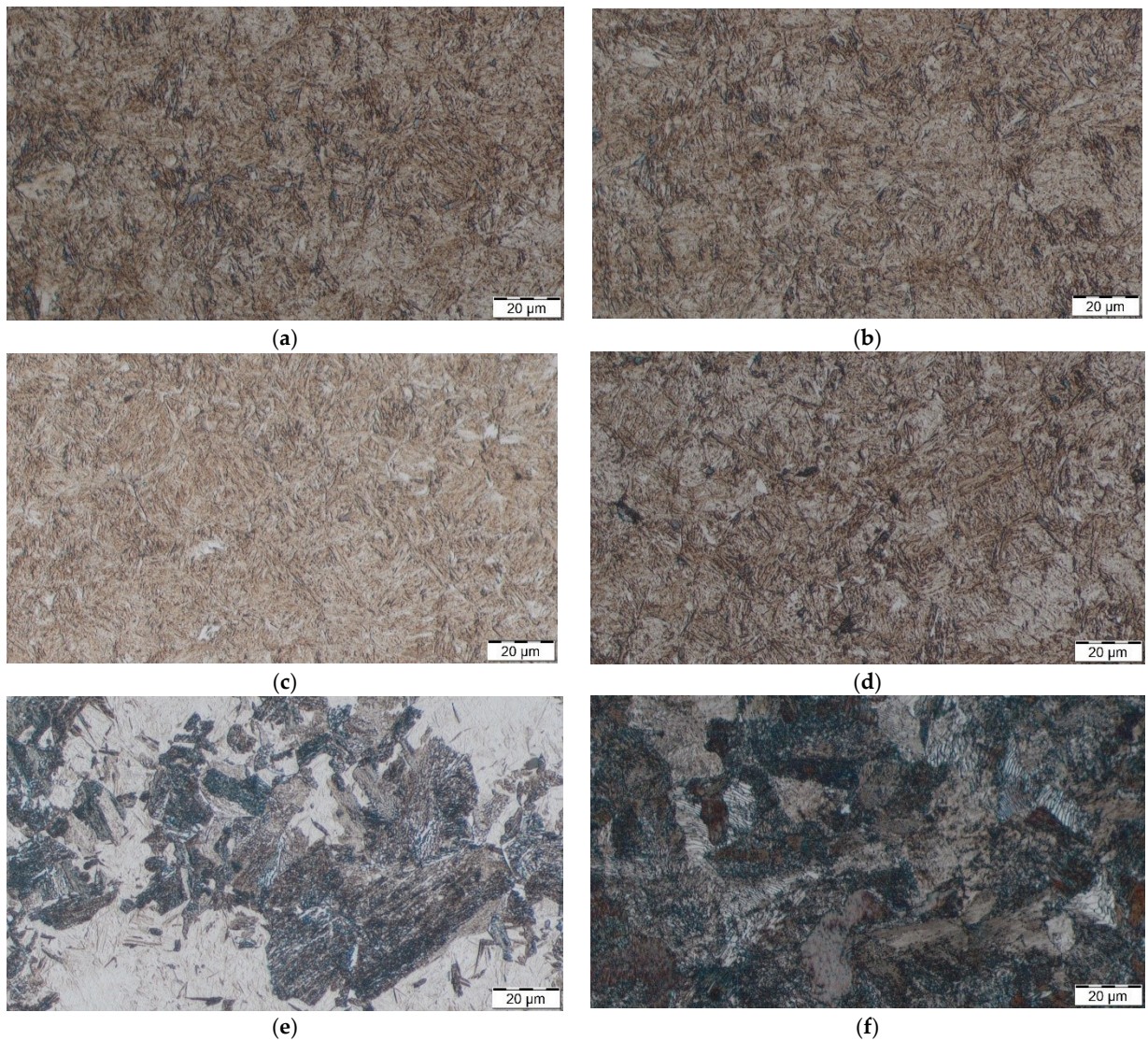

**Figure 5.** Optical micrographs of 55Cr3 cooled at (**a**) 100 °C/s, (**b**) 30 °C/s, (**c**) 20 °C/s, (**d**) 10 °C/s, (**e**) 1 °C/s, and (**f**) 0.1 °C/s.

### 3.4. Effect of Cooling Rates on the Microstructure and Hardness Properties

The overall microstructure for the cooling rate, 20 °C/s to 100 °C/s, for the three steels was martensite. The micrographs of 55Cr3 and 52CrMoV4, with a 10 °C/s cooling rate, presented mainly martensite and bainite microstructure. The bainite structure shown in Figure 5d was coarser compared to that observed in Figure 7d, showing the negligible presence of the bainite structure. Figures 5e and 7e indicated a significant percentage of bainite and martensite present, while Figure 6e had a pearlitic structure with the presence of pro-eutectoid ferrite. The percentage of bainite in Figure 5e was approximately equal to that of martensite, while martensite constituted a larger fraction in Figure 6e. Figures 5f and 7f showed pearlitic structures, while Figure 7f was pearlitic with proeutectoid ferrite. The details of the microstructure were confirmed with an SEM. The images of the SEM are presented in Figures 8–10. The microstructural characteristics were corroborated by determining the hardness measurements.

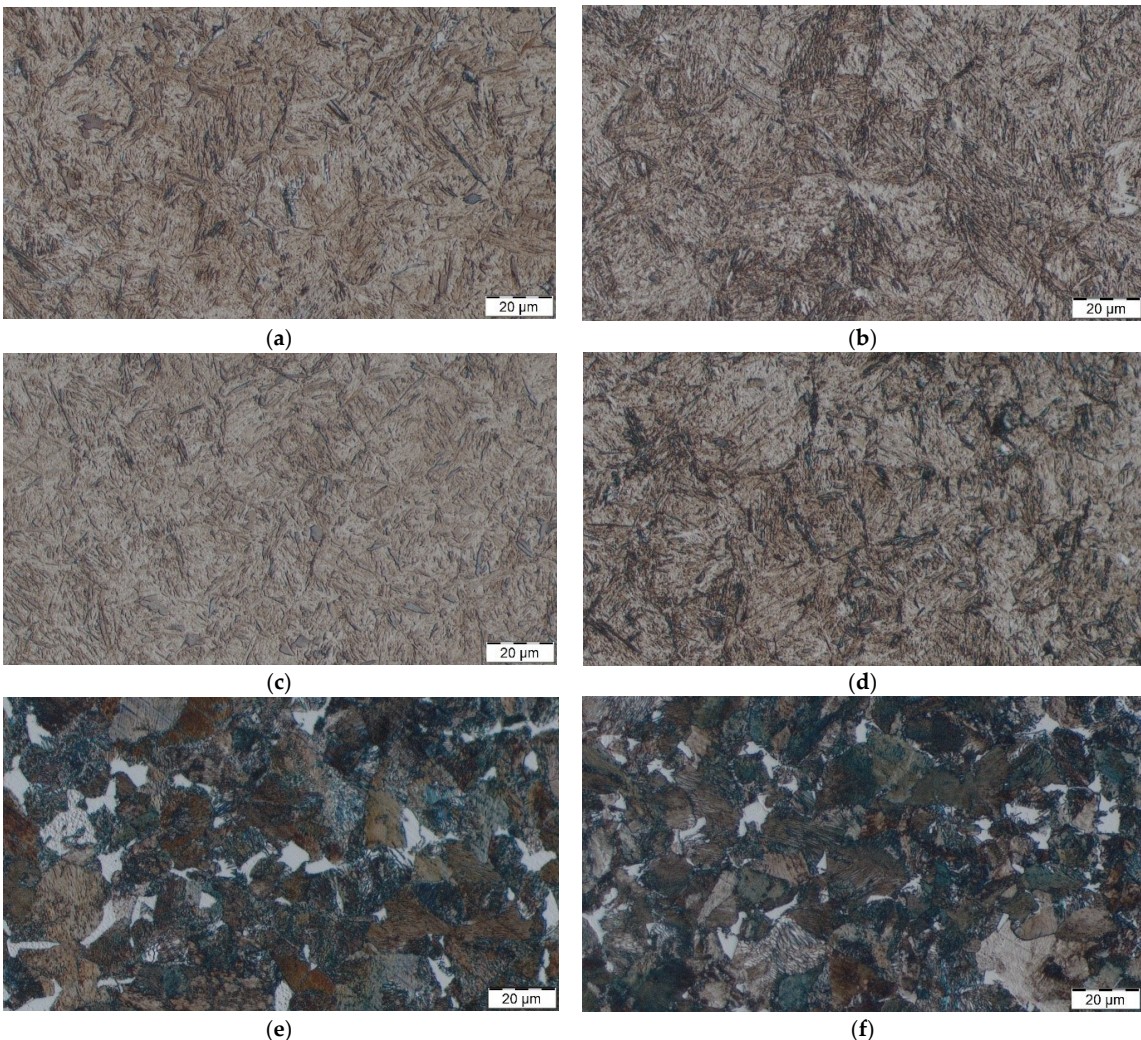

**Figure 6.** Optical micrographs of 54SiCr6 cooled at (**a**) 100 °C/s, (**b**) 30 °C/s, (**c**) 20 °C/s, (**d**) 10 °C/s, (**e**) 1 °C/s, and (**f**) 0.1 °C/s.

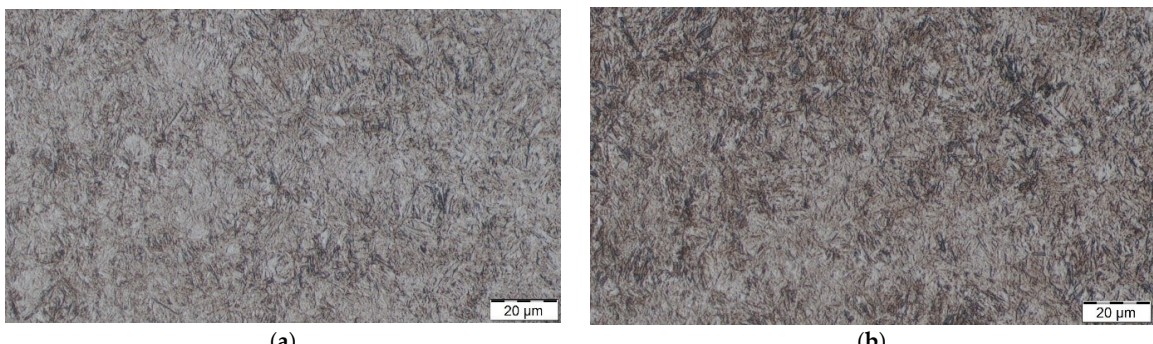

**Figure 7.** *Cont.*

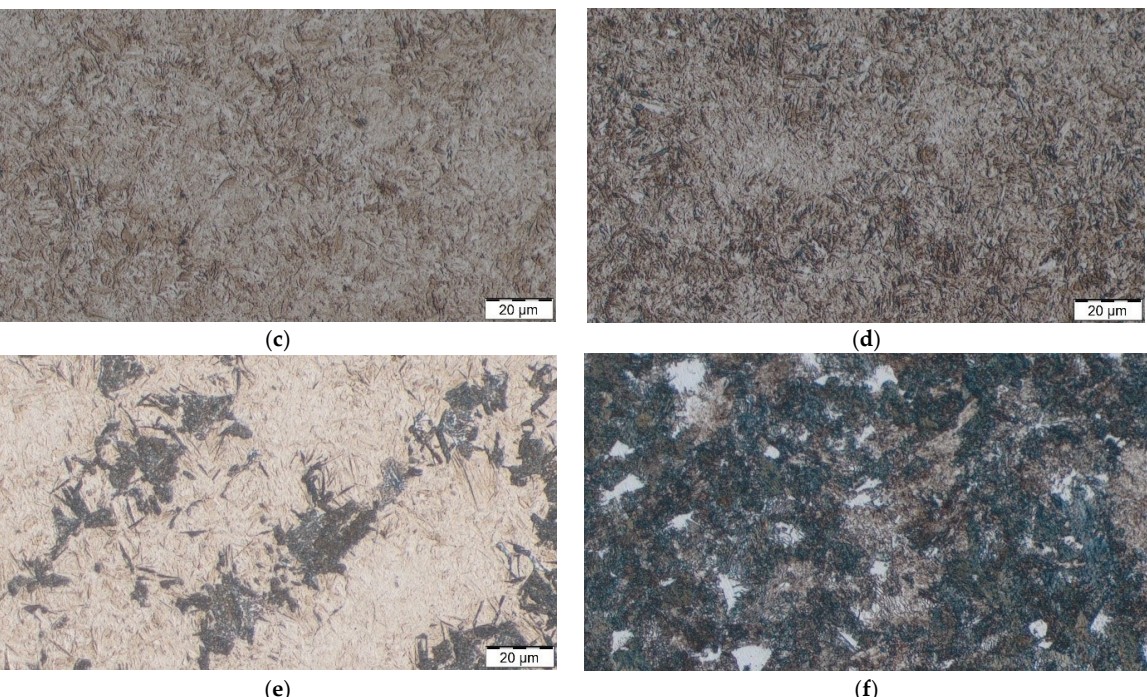

**Figure 7.** Optical micrographs of 52CrMoV4 cooled at (**a**) 100 °C/s, (**b**) 30 °C/s, (**c**) 20 °C/s, (**d**) 10 °C/s, (**e**) 1 °C/s, and (**f**) 0.1 °C/s.

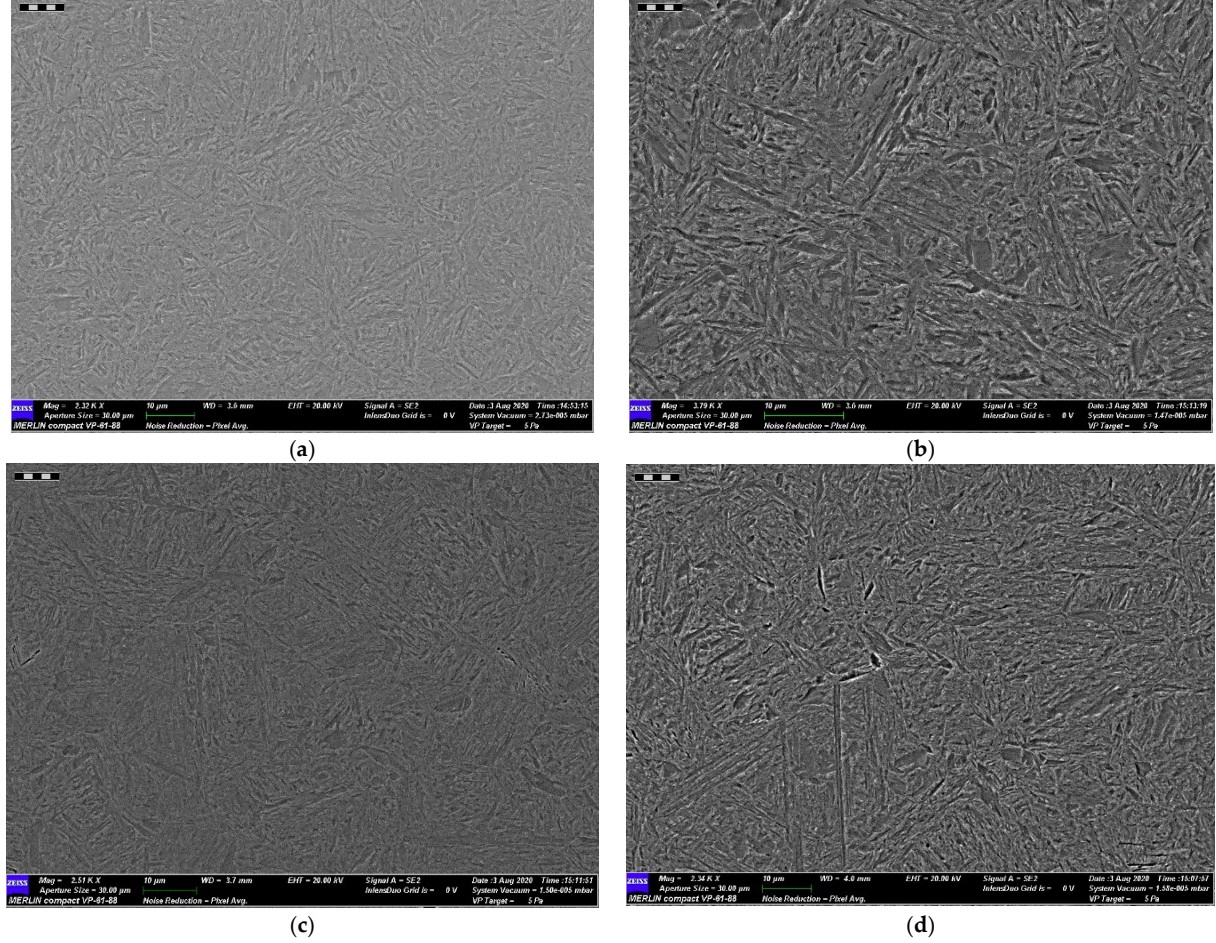

**Figure 8.** *Cont.*

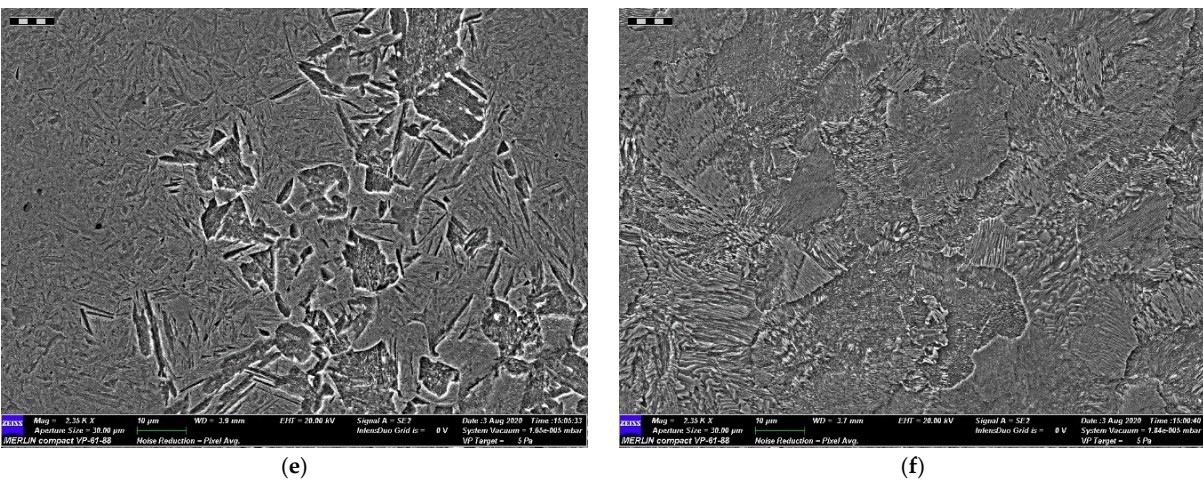

(**e**)　　　　　　　　　　　　　　　(**f**)

**Figure 8.** 55Cr3 SEM images of the microstructure cooled at (**a**) 100 °C/s, (**b**) 30 °C/s, (**c**).20 °C/s, (**d**) 10 °C/s, (**e**) 1 °C/s, and (**f**) 0.1 °C/s.

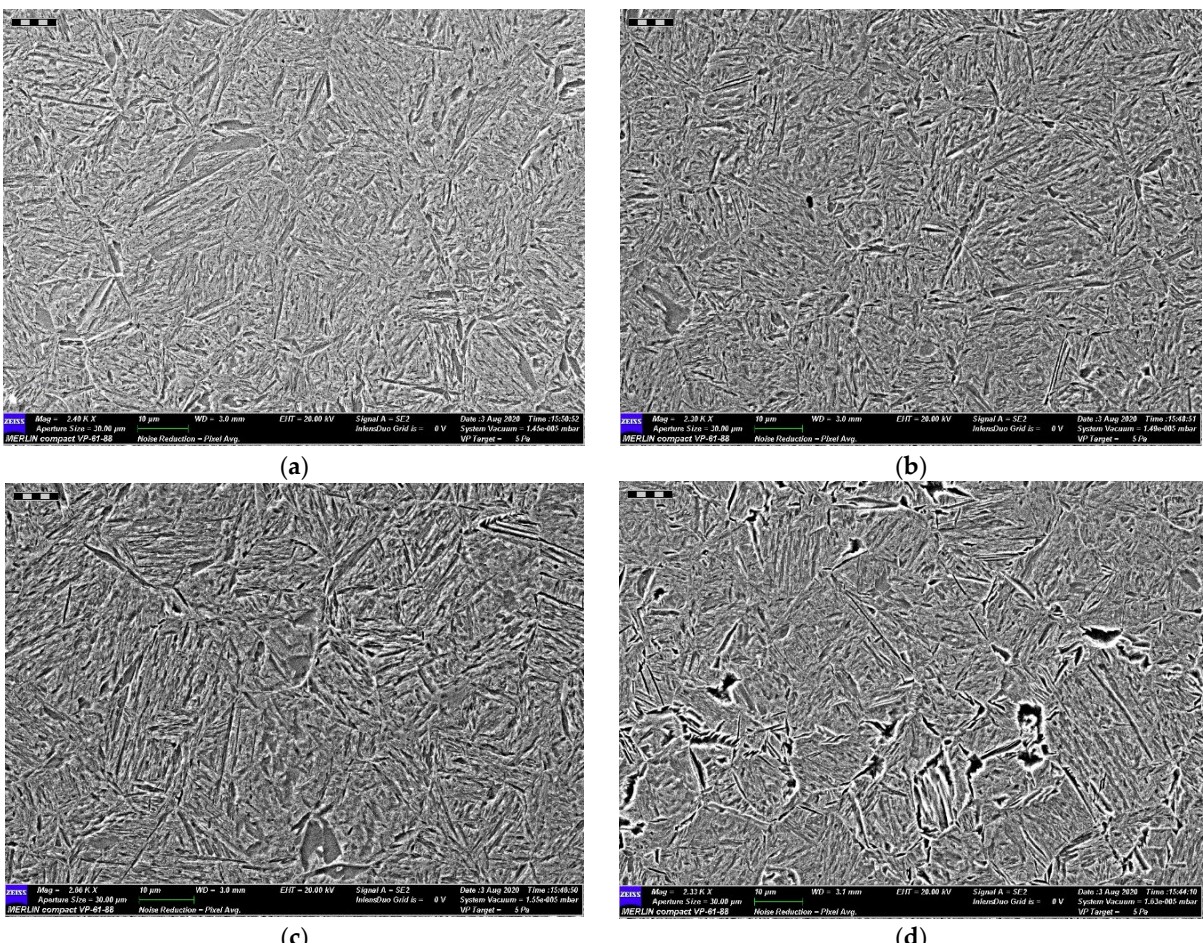

(**a**)　　　　　　　　　　　　　　　(**b**)

(**c**)　　　　　　　　　　　　　　　(**d**)

**Figure 9.** *Cont*.

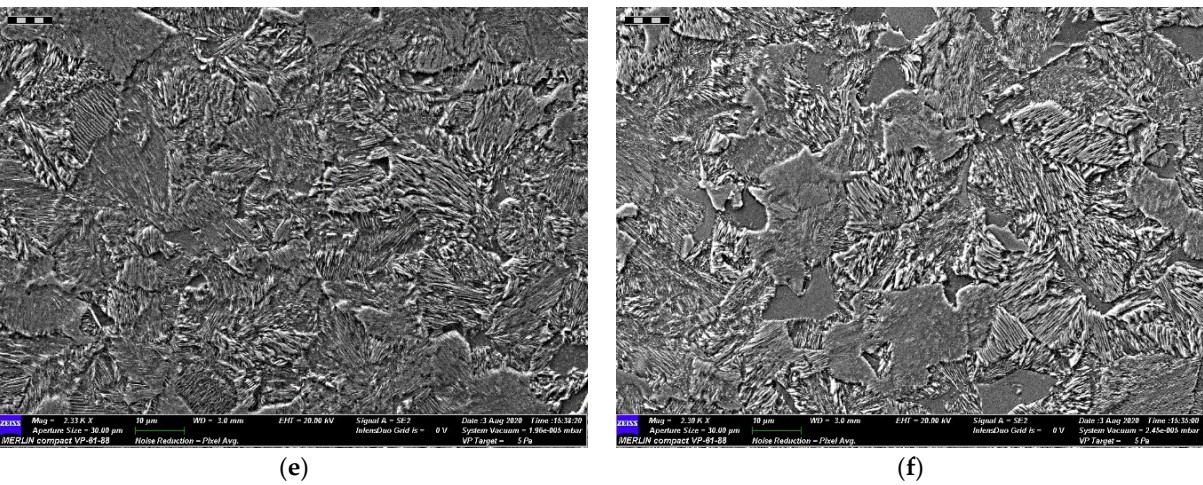

(**e**)                              (**f**)

**Figure 9.** 54SiCr6 SEM images of the microstructure cooled at (**a**) 100 °C/s, (**b**) 30 °C/s, (**c**) 20 °C/s, (**d**) 10 °C/s, (**e**) 1 °C/s, and (**f**) 0.1 °C/s.

(**a**)                              (**b**)

(**c**)                              (**d**)

**Figure 10.** *Cont.*

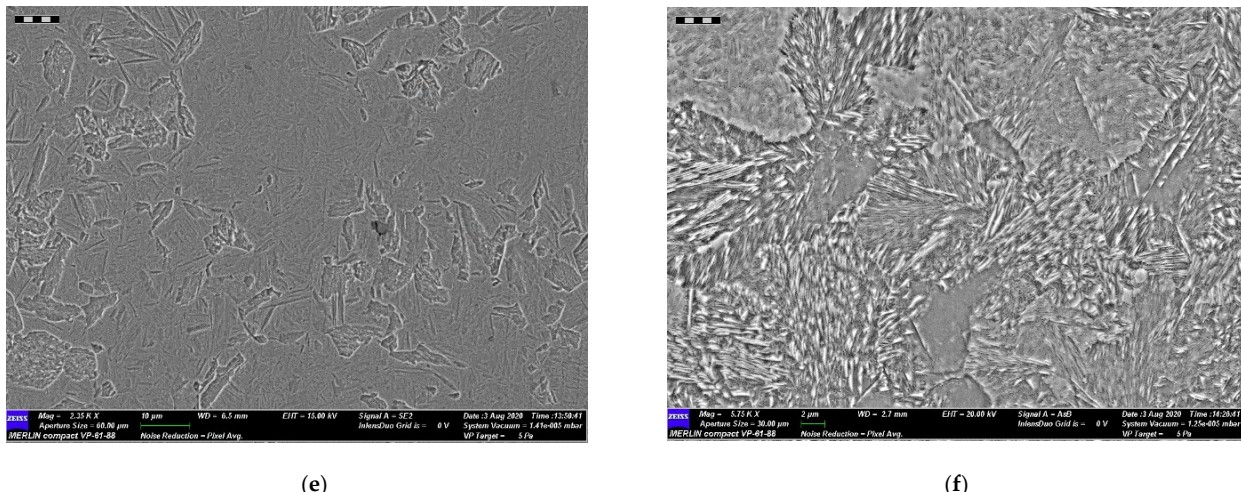

(**e**)                                        (**f**)

**Figure 10.** 52CrMoV4 SEM images of the microstructure cooled at (**a**) 100 °C/s, (**b**) 30 °C/s, (**c**) 20 °C/s, (**d**) 10 °C/s, (**e**) 1 °C/s, and (**f**) 0.1 °C/s.

To achieve adequate mechanical properties and microstructures, it is necessary to choose the required quenching media. The microstructural properties are further corroborated by the hardness measurements summarized in Table 4. Generally, with decreasing cooling rates, the hardness decreases.

**Table 4.** Hardness measurements determined at various cooling rates.

| | Micro-Vickers Hardness Tester | | |
|---|---|---|---|
| **Cooling Rate (°C/s)** | **55Cr3** | **54SiCr6** | **52CrMoV4** |
| 100 | 810 ± 36 | 802 ± 16 | 770 ± 11 |
| 30 | 827 ± 7 | 791 ± 9 | 770 ± 13 |
| 20 | 821 ± 10 | 791 ± 17 | 761 ± 41 |
| 10 | 820 ± 6 | 762 ± 17 | 739 ± 36 |
| 1 | 547 ± 126 | 336 ± 11 | 629 ± 137 |
| 0.1 | 299 ± 6 | 285 ± 11 | 296 ± 17 |

## 4. Conclusions

For the three spring steels, this analysis provided detailed critical temperatures for transformation. It is important to draw the following conclusions:

The results showed that to successfully optimise the heat treatment process parameters, detailed knowledge of the material is paramount. Generally, empirical heat treatment data are helpful for guidance; however, for optimisation purposes, the exact parameters are a requirement. Emanating from the dilatometric results, the optimum heating and quenching media can be predicted for each specific grade of spring steel.

54SiCr6 was the most hardenable, with the cooling rates ranging from 20 °C to 100 °C/s; however, it was the least hardenable, with slower cooling rates from 1 °C/s. In material hardenability, chemical composition plays an important role, but the cooling rate and critical transformation of austenite are equally crucial.

The data obtained from this study are adequate to optimize the heat treatment process, but to plot CCT for the three steels, additional dilatometer data are needed. The effect of tempering heat treatment on the microstucture that has been cooled at various temperatures will be measured as part of future work.

**Author Contributions:** Conceptualization, V.J.M. and J.W.v.d.M.; validation, N.L.V. All authors have read and agreed to the published version of the manuscript.

**Funding:** This research received no external funding.

**Institutional Review Board Statement:** Not applicable.

**Informed Consent Statement:** Not applicable.

**Conflicts of Interest:** The authors declare that they have no conflict of interest.

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
