# Peer review of "Determination of Critical Transformation Temperatures for the Optimisation of Spring Steel Heat Treatment Processes"

_metals, doi:10.3390/met11071014_

Round 1

Reviewer 1 Report

This manuscript is written as a technical report more than as a research article. A deeper discussion establishing the relationship among microstructures and dilatometry tests should be done to improve the quality of the article.

  • Line 35-36. The authors should mention the main differences in alloying elements.
  • Line 56-57. Andre´s equation should be shown the first time it was mentioned, line 46-47.
  • Line 42. Minor mistake. Please, remove the initial “The” in this sentence.
  • Line 102. Why the authors use a different heating rate (10 ºC/s) for the DSC tests and the dilatometry tests (5 ºC/s)? Could this affect the Ac1 and Ac3?
  • Another minor mistake in line 107. Please, remove “and” at the end of this sentence.
  • Figure 1. The authors should describe or cite the method used for measuring the critical temperatures.
  • Table 3. Could the authors report the Mf temperature alongside the Ms temperature? Does the cooling rate affect this?
  • Line 123. The authors should add some information regarding the initial microstructures of the steels. The authors discuss the dilatometry results in function of composition but the initial microstructure of the steels play a role too.
  • Figure 3. During the cooling some curves show  slope changes , the authors should measure these critical temperatures  and relate to the obtained microstructures.
  • Line 190. The authors should measure the amount of phases for each cooling rate. I strongly recommend discussing the morphology/amount of these phases in function of steel composition and transformation temperatures.

Reviewer 2 Report

The article investigates the transformation temperatures of three different spring steel grades and compares them with results obtained from empirical calculations.

The paper is well written, and the results are clearly presented.

Following are some comments to be addressed:

  • Line 80: how many samples per steel grade did you use for the dilatometric tests?
  • Table 2 would be much more readable switching the two entries:

Test sample\Elements

C

Si

Mn

P

S

55Cr3

54SiCr6

52CrMoV4

  • Lines 92: please explain in more detail metallographic preparation (what grinding step and polishing step did you use?)
  • Line 94: “2 % nitric reagent…” in what? If it is in alcohol please correct with 2% Nital and explain the composition
  • Regarding microhardness tests, please explain in much more detail how you performed the tests (i.e., load, time, and how many measures did you take per sample?)
  • There is something wrong with table 3. Please fix the issue
  • Line 128/132: there is some issue with the cross-reference since it is mentioned “Figure ??”.
  • Lines 134-137: You mention the fact that for 52CrMoV4 even after 45 minutes you did not observe austenite to be stabilized. It would be useful to completely stabilize austenite and compare the results with the other steels.
  • Figure 3: Insert steel grade name on each figure. Also, shift every curve in each figure in order to better read the data.
  • Figure 4: please change the y-axis range in order to better observe temperature changes within the three steel grades (e.g., from 200 °C to 300 °C)
  • Table 4: results in such notation are not correct. Please add ± sign. Moreover, why the uncertainty of 55Cr3 and 52CrMoV4 at 1 °C/s is so high? I suggest performing a higher number of microhardness tests to reduce the uncertainty.
  • Line 164: there is no such thing as “martensite with bainite microstructure”. Moreover, it is difficult to distinguish bainite from martensite by optical microscopy. SEM micrographs in figures from 8 to 10 must be improved in terms of brightness and contrast. It is better to change the SEM micrographs with ones at higher and constant magnification since it is difficult to distinguish from bainite and martensite.
  • Figure 5: caption error.
  • Line 172-173: please better explain the concept. I think you mean “corroborated” rather than “collaborated”.
  • There is no discussion section in the manuscript. It is not mandatory but I think it would be helpful to explain the obtained results.

Round 2

Reviewer 1 Report

The authors have answered the comments of the reviewer